# Energy filtering enables macromolecular MicroED data at sub-atomic resolution

Max T. B. Clabbers [1,2], Johan Hattne [1,2], Michael W. Martynowycz[1,2] & Tamir Gonen [1,2,3] ✉

High-resolution information is important for accurate structure modeling but is challenging to attain in macromolecular crystallography due to the rapid fading of diffracted intensities at increasing resolution. While direct electron detection essentially eliminates the read-out noise during MicroED data collection, other sources of noise remain and limit the measurement of faint high-resolution reflections. Inelastic scattering significantly contributes to noise, raising background levels and broadening diffraction peaks. We demonstrate a substantial improvement in signal-to-noise ratio by using energy filtering to remove inelastically scattered electrons. This strategy results in sub-atomic resolution MicroED data from proteinase K crystals, enabling the visualization of detailed structural features. Interestingly, reducing the noise further reveals diffuse scattering that may hold additional structural information. Our findings suggest that combining energy filtering and direct detection provides more accurate measurements at higher resolution, facilitating precise model refinement and improved insights into protein structure and function.

At atomic resolution, commonly defined as beyond 1.2 Å, individual atoms are fully resolved, revealing an accurate model of the underlying structure[1]. Unfortunately, noise in the measurement often makes it difficult to obtain such information from macromolecular crystals, as the mean diffracted intensity decreases rapidly at higher resolution. This cannot be circumvented by simply increasing the fluence as this leads to an increase in detrimental radiation damage, limiting the total structural information that can be recovered from each crystal. This necessitates the use of a low total fluence in macromolecular data collection. The expected signal-to-noise ratio is therefore poor, limiting the attainable resolution. Sufficiently strong data can still be retrieved by microcrystal electron diffraction (MicroED)[2], supported by the use of fast and highly sensitive cameras that are effective at low flux densities[2–4]. Recently, we reported a substantial improvement in data quality by recording data using electron counting on direct electron detectors[5]. The improved accuracy and resolution enabled ab initio phasing using a fragment-based approach[5] and allowed the identification of hydrogen bond networks and the protonation state of atoms[6]. However, this was demonstrated for triclinic lysozyme, which

is relatively small and forms crystals with low solvent content, while proteinase K, which is much larger and contains more solvent, did not reach similarly high resolution. Electron counting detectors do not eliminate all noise, and further optimization of the experimental setup and data collection strategies is needed to routinely obtain more high-resolution information.

Inelastic scattering is a major factor limiting the signal-to-noise ratio[7,8]. Upon an inelastic interaction, electrons lose a small fraction of their energy and are no longer coherent. This contributes to increased background noise and a broadening of the Bragg peaks, thereby affecting the kinematic diffraction signal. This poses a significant challenge, as inelastic events are 3–4 times more likely than elastic scattering in biological specimens[8,9]. Whereas lattice filtering or subtracting a smooth radial background can partially correct for this[10,11], post-processing does not reduce the noise inherently present in the measurement and cannot recover weak high-resolution reflections that are at or beneath the noise level. Removing inelastically scattered electrons experimentally based on their energy loss is therefore preferable. Previous reports assessed the effects of energy filtering on

[1]Howard Hughes Medical Institute, University of California, Los Angeles, CA 90095, USA. [2]Department of Biological Chemistry, University of California, Los Angeles, CA 90095, USA. [3]Department of Physiology, University of California, Los Angeles, CA 90095, USA. ✉e-mail: tgonen@g.ucla.edu

macromolecular diffraction data, demonstrating a significant reduction in background noise and sharper spots, resulting in improved data quality statistics[3,12,13]. Owing to detector geometry, the resolution was not improved even though the signal-to-noise ratio increased substantially[13].

Here, we show that combining energy filtering with the improved accuracy and sensitivity of direct electron detection enhances data quality and resolution in MicroED. Two important considerations to this approach are the total fluence that can be withstood by the crystals, and the flux tolerated by the camera before coincidence loss prevents an accurate representation of the counts. The diffraction signal, therefore, is correspondingly weaker. To mitigate this, we use a protocol that continuously rotates the crystals slower, uses the same total fluence spread over a smaller 20° sweep, and utilizes the energy filter to further improve the signal-to-noise ratio. We apply this strategy to collect electron-counted and energy-filtered MicroED data from crystalline lamellae of proteinase K that are machined to an optimal thickness of ~300 nm[14].

## Results

Initial diffraction images were taken to assess lamellae quality and achievable resolution (Fig. 1). Diffraction spots extended to the edge of the detector beyond 1.0 Å resolution, showing less noise, reduced background, and sharper Bragg peaks compared to unfiltered data (Supplementary Figs. 1 and 2)[5,15]. Interestingly, energy filtering revealed diffuse scattering from the bulk solvent that is largely obscured in unfiltered data (Fig. 1)[5,15]. Continuous rotation MicroED data were collected from the best crystals, showing high-quality information and spots extending to ~1 Å resolution (Supplementary Fig. 3). Individual datasets were processed and showed significant information ranging from 1.13 to 1.06 Å resolution (Supplementary Fig. 4). Not all lamellae diffracted to the same resolution depending on crystallinity, quality of the lamellae, and lattice orientation.

Data from 17 crystals were merged and truncated at 1.09 Å resolution to ensure high multiplicity and complete sampling of reciprocal space (Supplementary Table 1). Previously, a maximum resolution of 1.4 Å was achieved from similar lamellae without the use of energy filtration[15]. Intensity statistics show a clear improvement in attainable resolution and a more gradual decrease in crystallographic quality indicators compared to unfiltered data previously reported using 63° sweeps on the same experimental setup without an energy filter (Fig. 2A, Supplementary Table 2)[15]. Additionally, we collected unfiltered data using the same 20° sweep strategy described here, which did not yield any improvement in data quality over the previous unfiltered data from the larger sweeps (Supplementary Table 2). Importantly, improving the resolution more than doubled the number of unique reflections used for structure refinement (Supplementary Tables 1 and 2). At atomic resolution, the electrostatic potential map shows highly resolved features, enabling accurate interpretation of the structural model and visualization of hydrogen atoms (Fig. 2B).

## Discussion

We demonstrate that optimizing MicroED data collection by integrating an energy filter with direct electron detection cooperatively increases data quality and resolution by mitigating the effects of inelastic scattering. The filtered data exhibited reduced background noise, sharper spots, and an improved signal-to-noise ratio. Although inelastically scattered electrons with minimal energy loss may contribute some useful signal, their phase shift reduces structural coherence, and those with significant energy loss primarily add to background noise. The most coherent electron contributions can be isolated using energy filtering, thus enhancing the signal and data quality. Better and higher resolution data provide more detailed insight into protein structure and function and may prove useful in visualization of hydrogen atoms and hydrogen bonding networks[6], as well as charge distribution[3,10]. With this improved data collection

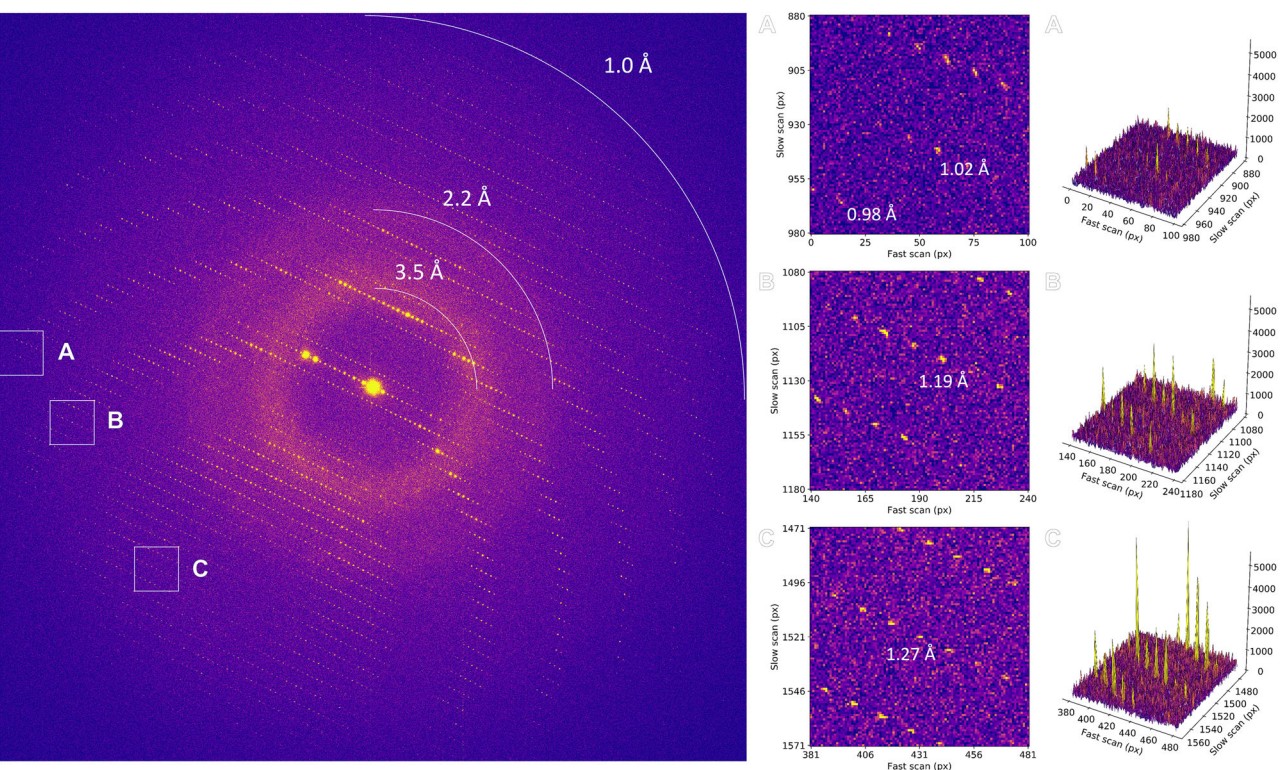

**Fig. 1 | Sub-atomic resolution energy-filtered MicroED data.** Energy-filtered diffraction pattern of a stationary proteinase K lamella was recorded over a 420 s exposure at a total fluence of 0.84 e⁻/Å². Highlighted areas are magnified in the right panels, and the corresponding peak profiles are plotted.

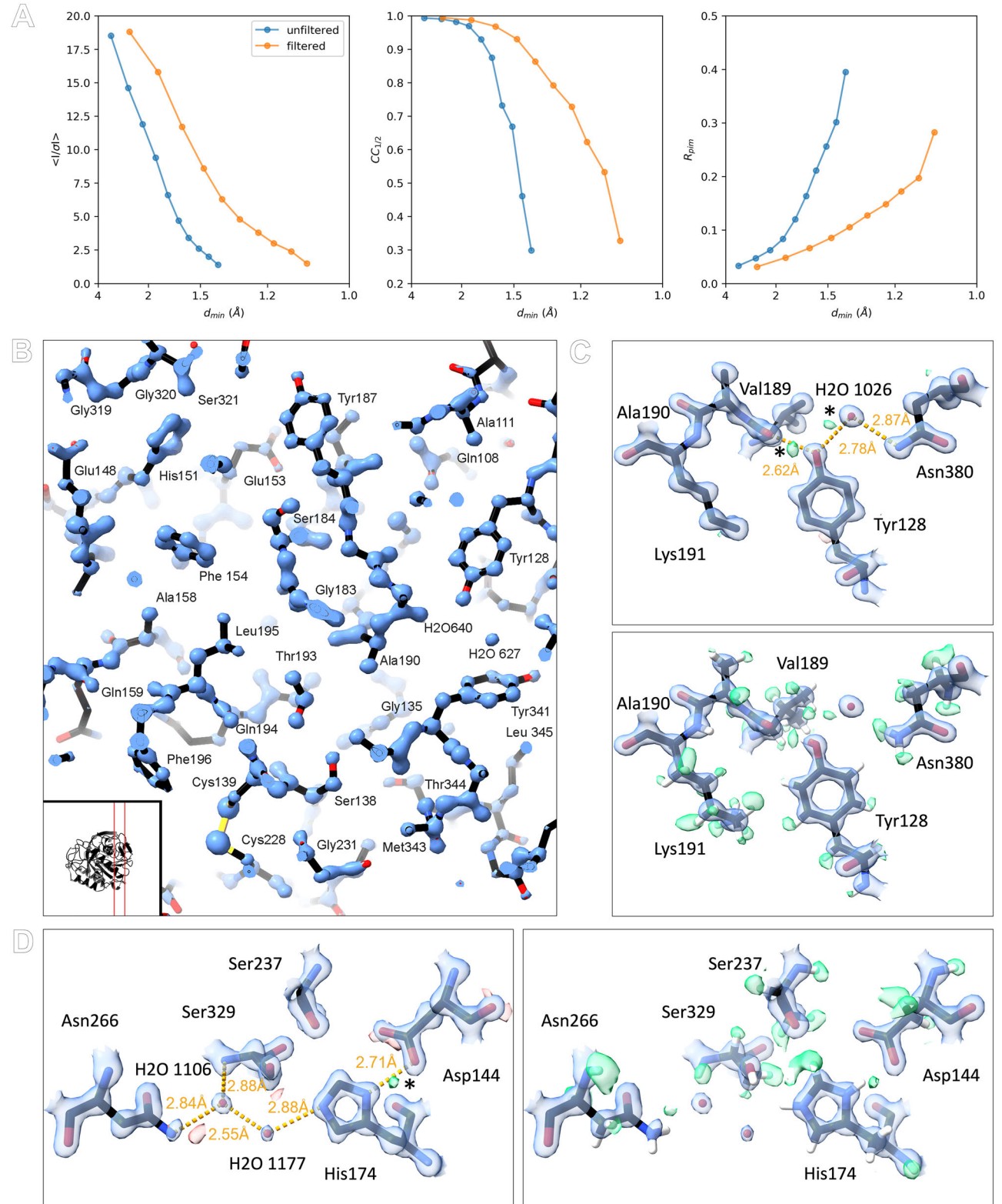

**Fig. 2 | Energy-filtered MicroED data show improved statistics and high-quality maps. A** Comparison of intensity statistics for filtered and unfiltered MicroED data, featuring from left to right: $I/\sigma I$, $CC_{1/2}$, and $R_{pim}$. **B** Slice through the structural model showing the electrostatic potential map, the location of the slice in the structure is indicated in the inset. **C** Maps and hydrogen omit maps are shown for Tyr128. **D** Maps and hydrogen omit maps for active site residues and waters. Electrostatic potential 2mFo−DFc maps are shown in blue and contoured at $4\sigma$ (**B**), and $2.2\sigma$ (**C**, **D**). Difference mFo−DFc maps are shown in green and red for positive and negative density contoured at $3\sigma$ (**C**, **D**). Hydrogen omit maps are shown in green and contoured at $2.3\sigma$ (**C**, **D**). Difference peaks marked with an asterisk indicate potential hydrogen atoms that have not been included in riding positions prior to map calculations.

approach, the attainable resolution for proteinase K is similar to reported proteinase K structures by X-ray crystallography[16,17], indicating that as technologies improve MicroED can rival more established structural biology methods. At comparable resolution, the atomic positions are more clearly resolved in the electron density maps, highlighting the need for more accurate electron scattering factors and improved modeling of the electrostatic potential distribution[3,10].

Diffraction spots extended beyond 1.0 Å resolution on individual frames, suggesting that there is still room for improvement. Our optimized protocol, which utilizes small 20° sweeps and therefore allows for an increased flux per slice through reciprocal space, may have further contributed to the improved signal-to-noise ratio. Interestingly, the smaller sweeps alone did not improve the unfiltered data compared to the larger 63° sweeps used previously, indicating that the improvements in data quality and resolution can mainly be attributed to the removal of the inelastic scattering contributions. The decay in mean diffracted intensity and subsequent loss of high-resolution information can be expected to occur identically in either of the data collection strategies as both resulted in a similar dose[18]. A data collection strategy that combines a larger number of datasets[3], or a serial approach using individual snapshots[11], may further improve data quality and resolution. Furthermore, the possibility to separate inelastic scattering from the signal is unique for electrons. It uncovers diffuse scattering, which can originate from protein disorder or dynamics, providing additional information that improves accurate modeling and refinement[19]. As energy filtering removes a major obstacle to better data quality, other sources of noise that remain can be addressed more effectively and potentially lead to further improved accuracy in macromolecular electron crystallography.

## Methods

### Crystallization
Proteinase K powder (*Engyodontium album*) was purchased from Sigma Aldrich and used without further purification. Crystals were grown in batch by dissolving 40 mg/ml proteinase K in 20 mM MES-NaOH pH 6.5. The protein solution was mixed at a 1:1 ratio with a precipitant solution of 0.5 M NaNO$_3$, 0.1 M CaCl$_2$, 0.1 M MES-NaOH pH 6.5. The mixture was incubated at 4 °C. Microcrystals with dimensions ranging between 7 and 12 μm grew within 24 h.

### Sample preparation
Standard holey carbon electron microscopy grids (Quantifoil, Cu 200 mesh, R2/2) were glow discharged for 30 s at 15 mA on the negative setting. Samples were prepared using a Leica GP2 vitrification device set at 4 °C and 90% humidity. For each sample, 3 μl of crystal solution was deposited onto the grid, incubated for 10 s, and any excess liquid was blotted away from the backside. Next, the sample was soaked for 30 s with 3 μl cryoprotectant solution containing 30% glycerol, 250 mM NaNO$_3$, 50 mM CaCl$_2$, 60 mM MES-NaOH pH 6.5. After incubation, any excess solution was blotted away using filter paper for a second time. Immediately after, the grid was rapidly vitrified by plunging it into liquid ethane. Grids were stored in a liquid nitrogen dewar prior to further use.

### Focused ion beam milling
Grids were loaded onto a Helios Hydra 5 CX dual-beam plasma FIB/SEM (Thermo Fisher Scientific). Prior to milling, grids were coated with a thin protective layer of platinum for 45 s using the gas injection system. Microcrystals of proteinase K were machined using a stepwise protocol to an optimal thickness of approximately 300 nm using a 30 kV Argon plasma ion beam. Coarse milling steps were performed using a 2.0 nA current to a thickness of approximately 3 μm. Finer milling steps at 0.2 nA were used to thin the lamellae to 600 nm. Final polishing steps were performed at 60 pA down to 300 nm thickness and 5 μm in width. Grids were cryo-transferred immediately after to

the TEM for data collection. Grids were rotated by 90° relative to the milling direction such that the rotation axis on the microscope is perpendicular to the milling direction.

### Hardware setup
Data were collected on a Titan Krios G3i TEM (Thermo Fisher Scientific) equipped with an X-FEG operated at an acceleration voltage of 300 kV, a post-column Selectris energy filter, and a Falcon 4i direct electron detector. The microscope was aligned for low flux density conditions using the 50 μm C2 aperture, spot size 11, and gun lens setting 8 for a less bright but more coherent illumination. A parallel electron beam of 10 μm diameter was used for data collection. The flux density at these conditions is approximately 0.002 e$^-$/Å$^2$/s. The energy spread of the emitted electrons was characterized as $\Delta E = 0.834 \pm 0.006$ eV at FWHM. The zero-loss peak of the energy filter was first centered in imaging mode. In our system, there is an offset in the position of the zero-loss peak when switching between imaging and diffraction modes. Therefore, the energy filter slit was carefully offset manually in defocused diffraction mode to align it with the selected area (SA) aperture used for diffraction data collection.

### Data collection
Electron-counted and energy-filtered MicroED data were collected using the continuous rotation method. Diffraction data were collected using a 150 μm SA aperture, corresponding to a beam diameter of ~3.5 μm at the sample plane as defined by the aperture. The energy filter was tuned to pass electrons with energy losses less than 10 eV, with the zero-loss peak centered in defocused diffraction. The effective sample-to-detector distance was calibrated at 1402 mm using a standard evaporated aluminum grid (Ted Pella). Crystals were rotated at a slow angular increment of 0.0476 °/s covering a total tilt range of 20.0°. Data were collected over 420 s exposures at a total fluence of ~0.84 e$^-$/Å$^2$. Equivalent dose values were calculated using the EMED subprogram of RADDOSE-3D[20]. Diffraction stills were taken using the same settings without any stage rotation from stationary crystals. No beam stop post-energy filter was used. Even without a beam stop, the low-resolution reflections close to the center, where most of the inelastic scattering normally accumulates, are in sharp contrast to the low background. Data were recorded on a Falcon 4i direct electron detector in electron counting mode operating at an internal frame rate of ~320 Hz. The proactive dose protector was manually disabled. Raw data were written in electron event representation (EER) format with an effective readout speed of ~308 frames per second.

### Data processing
Individual MicroED datasets in EER format were binned by two and converted to SMV format using the MicroED tools (available at https://cryoem.ucla.edu/downloads), after summing batches of 308 frames and applying post-counting gain corrections. Individual MicroED datasets were processed using XDS[21]. The sample-to-detector distance was not explicitly refined during data processing. Data were integrated up to a cross correlation between two random half sets that was still significant at the 0.1% level[22]. All frames were used during data processing. Individual datasets were analyzed and merged using XSCALE[21] and XSCALE_ISOCLUSTER[23]. The merged data were truncated at a mean $I/\sigma I \geq 1.0$ and a CC$_{1/2}$ of 33.0% in the highest resolution shell. Data were merged using Aimless[24].

### Structure solution and refinement
The structure was phased by molecular replacement using electron scattering factors in Phaser[25]. The model was inspected and built using Coot[26]. Two calcium ions and one nitrate were placed, and a total of 21 residues were modeled in alternate conformations. The structure was refined using REFMAC5[27] and Phenix.refine[28]. Refinement was done using electron scattering factors and individual anisotropic *B*-factors

for all atoms except hydrogens. Hydrogen omit maps were calculated using REFMAC5. Difference peaks higher than $2\sigma$ and $3\sigma$ in the omit map within less than 1.0 Å from any known riding positions were identified as potential hydrogen atoms.

### Data comparison

The energy-filtered MicroED data were compared to two unfiltered datasets that were collected using the same experimental setup: the first set was obtained by merging unfiltered data from five proteinase K lamellae using 63.0° sweeps at a total fluence of ~1.0 e⁻/Å² (3.7 MGy) and was previously reported in Martynowycz et al.[15]. The second set merged unfiltered data from 12 lamellae using the same 20.0° sweep data collection strategy as used for the filtered MicroED data, where each dataset was recorded using a total fluence of ~0.84 e⁻/Å² (3.1 MGy) with the energy filter slit retracted. The merged data were truncated at 1.4 Å resolution to enable an equal comparison of the intensity and model statistics.

### Reporting summary

Further information on research design is available in the Nature Portfolio Reporting Summary linked to this article.

## Data availability

The data that support this study are available from the corresponding authors upon request. The EM map has been deposited in the Electron Microscopy Data Bank (EMDB) under accession code EMD-46871 (Proteinase K). The atomic coordinates and structure factors have been deposited in the Protein Data Bank (PDB) under accession code 9DHO (Proteinase K).

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

## Acknowledgements

This study was supported by the National Institutes of Health (P41GM136508), the Department of Defense (HDTRA1-21-1-0004), and the Howard Hughes Medical Institute. Portions of this research or paper completion were developed with funding from the Department of Defense MCDC-2202-002. Effort sponsored by the U.S. Government under Other Transaction number W15QKN-16-9-1002 between the MCDC, and the Government. The US Government is authorized to reproduce and distribute reprints for Governmental purposes, notwithstanding any copyright notation thereon. The views and conclusions contained herein are those of the authors and should not be interpreted as necessarily representing the official policies or endorsements, either expressed or implied, of the U.S. Government. The PAH shall flow down these requirements to its sub awardees, at all tiers. M.W.M is currently at SUNY Buffalo.

## Author contributions

T.G. and M.T.B.C. designed the research. M.T.B.C. prepared samples and collected data. M.T.B.C., J.H., and M.W.M. analyzed the data. The paper was written by M.T.B.C. and T.G. with input from all authors. T.G. managed the project.

## Competing interests

The authors declare no competing interests.
