## [Transparent Peer Review file · Nature Communications]

Energy filtering enables macromolecular MicroED data at sub-atomic resolution

Corresponding Author: Professor Tamir Gonen

Version 0:

Reviewer comments:

Reviewer #1

(Remarks to the Author)

The article, titled "Energy filtering enables macromolecular MicroED data at sub-atomic resolution," demonstrates how energy filtering combined with electron counting significantly improves the signal-to-noise ratio in microcrystal electron diffraction (MicroED). This improvement builds upon the authors previous result with lysozyme and enables sub-atomic resolution data collection on proteinase K, allowing for detailed structure modeling of the large macromolecule. The paper highlights how this method reduces background noise, enhances data quality, and reveals diffuse scattering phenomena. The authors also address issues of inelastic scattering and coincidence losses. Ultimately, this technique offers new insights into protein structure determination and facilitates more accurate model refinements. I recommend the paper be accepted for publication.

Reviewer #2

(Remarks to the Author)

Summary

The manuscript describes the use of an energy selecting filter combined with electron counting detector for the collection of electron diffraction data from lamellae of proteinase K. This builds on previous work by the authors demonstrating the use of electron counting detectors for electron diffraction. The manuscript argues and concludes that the use of the energy filter improves the overall resolution of the data by filtering out the inelastically scattered electrons. Evidence for this is provided in the form of example diffraction images with specific reflections highlighted with pixel intensity plots. The manuscript also provides multiple processing statistics from a number of data collections using the energy filter. Example electrostatic potential maps are provided to demonstrate the quality of the data and difference maps are provided to demonstrate that the data includes density for hydrogen positions. There is a limited comparison of the energy filtered data with the unfiltered data, particularly comparing the data quality indicators to those from previously published work (Martynowycz, M. W. et al. Nat. Commun. 14, 1086 (2023)).

The authors refer to work by Yonekura et al. (Yonekura, K., Ishikawa, T. & Maki-Yonekura, S. A new cryo-EM system for electron 3D crystallography by eEFD. J. Struct. Biol. 206, 243–253 (2019)) and state that their work does not demonstrate any increase in resolution due to the use of energy filter. This is true because the resolution of their experiments was limited by detector geometry. However, the data quality statistics indicate clearly that the signal to noise of the data is substantially improved (by about a factor of 3 in both their test cases at the highest measured resolution) where energy filtering is employed. For completeness, this point should be mentioned by the authors of this manuscript.

While the manuscript demonstrates an increase in attainable resolution through the use of an energy filter combined with electron counting detector, it demonstrates this on one well diffracting system and on measurements recorded from different sample crystals. Overall, in its current state the manuscript lacks the sufficient detail and a robust comparison with non-energy filtered data to support the conclusions made. Specifically, it is unclear which 'unfiltered' data sets were used for the comparison and precisely how they were prepared, which lamella thicknesses were used, how many crystals were merged, and so on. At very least a side-by-side comparison table between unfiltered and filtered data should be included detailing

this information to allow the reader to judge their interpretation. An ideal experimental comparison would have been to record filtered and unfiltered from the same lamella, thereby reducing impact of crystal-to-crystal variation. Was this considered and tried? The reviewers suggest that this may be a more rigorous demonstration of any observed gains in signal to noise.

The manuscript is of interest to the emerging community of electron diffraction users and facilities and represents a potential incremental but important improvement to data quality. To support their conclusions, we suggest that the authors consider the suggestions made above and also consider a new set of experimental data collections.

Detailed Comments

- Are the distances appropriate for the positive densities in figure 2 to be representative of protons?
- Given that some fraction of the inelastically scattered electrons contribute to the Bragg peaks (i.e. elastic event followed by inelastic event), and contribute to measurable and potentially useful signal, what impact does this have on the conclusions drawn?
- What were the last diffraction weighted doses? The manuscript refers to low electron fluence, but it would be good to understand the impact as dose. These can now be easily calculated using Raddose-ED. Particularly to be able to compare the required doses needed for filtered Vs un-filtered for equivalent diffraction quality.
- The extended data in the manuscript depicts diffraction using 2 sizes of energy selecting slit but no comment or comparison is made on the effect of using the narrower slit.
- The manuscript only demonstrates a single crystal system. The manuscript would be strengthened by the demonstration of the technique on a different crystal system where the unit cell size, spacegroup and protein mass are reasonably different.
- What was the distribution of the combined datasets in terms of resolution range, were all the images used? Were all the images used from each individual dataset?
- What do the statistics look like for an equivalent 17 merged datasets from non-energy filtered conditions in the same system?
- Figure 2A. The comparison of intensity statistics for filtered Vs unfiltered. Unclear if these are merged or single data sets.
- Extended figure 4 – CC1/2 please add dotted line denoting the resolution cut-off on for CC1/2 axis.
- The comparable data between filtered and unfiltered in the manuscript is limited, it extends only to diffraction images and a single comparison of intensity, CC1/2 and Rpim statistics and it is not clear how reproducible or robust these are.
- Please include unfiltered data in extended data table 1 so that comparison of statistics can be made. E.g. there are no values for unfiltered obs. or unique obs. and so the associated statement in the text cannot be backed up.
- Can the background signal be quantified for energy filtered vs unfiltered?
- It is not clear what extended figure 2 offers in addition to figure 1. Which energy filter was used to generate figure 1?
- Figure 2. A. “Comparison of intensity statics for ...” I think should read “Comparison of intensity statistics for ...”
- Did the refined detector distance correlate with the calibrated distance and was this the same for all datasets?
- What were the approximate/average lamella dimensions? Were the lamellae narrower than the parallel electron beam?
- The manuscript should include data processing scripts in the supplementary information.

Reviewer #3

(Remarks to the Author)
Please see attached PDF file

Reviewer #4

(Remarks to the Author)
I co-reviewed this manuscript with one of the reviewers who provided the listed reports. This is part of the Nature Communications initiative to facilitate training in peer review and to provide appropriate recognition for Early Career Researchers who co-review manuscripts.

Version 1:

Reviewer comments:

Reviewer #2

(Remarks to the Author)
We would to thank the authors for their work in amending the manuscript.

The manuscript has been substantially improved with the introduction of a clearer side-by-side comparison of the unfiltered and filtered data sets. Our overriding concern still is that the significantly wider sweeps of data per crystal (63 degs. vs 20 degs.) may well mean the unfiltered data were more impacted by beam damage. i.e. the resolution at the start of the data collection may have been higher than that at the end of data collection and therefore higher than the final resolution cut off. All of the other concerns have been addressed.

We suggest that the authors check the data reduction scale factors of both filtered and unfiltered data sets and quantify the beam damage effects (looking at the scaling B factors for example, or loss of mean intensity at a given resolution) for both filtered and unfiltered sets. This would be to confirm that beam damage in the unfiltered data sets does not contribute substantially to the differing data quality of filtered vs unfiltered data.

Comments

1. In the Data Collection section, it would be very helpful to explicitly include the equivalent data collection parameters used for the original unfiltered data sets. These could possibly be included in a Table for clarity.
2. Figure 1 legend. "... over a 420 s expose at a total ..." should read "... over a 420 s exposure at a total ..."

Reviewer #3

(Remarks to the Author)

I appreciate the authors' efforts in addressing the reviewers' comments and improving the manuscript. I am pleased with the revisions and fully support its publication in Nature Communications.

I would like to reiterate that the improved structure determination is not solely due to the use of the energy filter, but also to the updated small-wedge data collection strategy. I believe the higher fluence per frame contributed to the improved signal-to-noise ratio as well (newly added Extended Data Table 2). I appreciate that the authors included a statement on this in the revised manuscript.

Reviewer #4

(Remarks to the Author)

Version 2:

Reviewer comments:

Reviewer #2

(Remarks to the Author)

I would to thank the authors for their work in addressing reviewers comments. The manuscript can now be submitted as is and I look forward to seeing in print.

Reviewer #3

(Remarks to the Author)

I am satisfied with the revised manuscript.

Reviewer #4

(Remarks to the Author)

Response to Reviewer Comments:

We thank the reviewers for their interest and useful suggestions. We addressed all comments in the revised version of our manuscript. Here, we reply to each of the comments and describe the changes that we have made to accommodate them. Our responses are highlighted in blue.

Reviewer #1 (Remarks to the Author):

The article, titled "Energy filtering enables macromolecular MicroED data at sub-atomic resolution," demonstrates how energy filtering combined with electron counting significantly improves the signal-to-noise ratio in microcrystal electron diffraction (MicroED). This improvement builds upon the authors previous result with lysozyme and enables sub-atomic resolution data collection on proteinase K, allowing for detailed structure modeling of the large macromolecule. The paper highlights how this method reduces background noise, enhances data quality, and reveals diffuse scattering phenomena. The authors also address issues of inelastic scattering and coincidence losses. Ultimately, this technique offers new insights into protein structure determination and facilitates more accurate model refinements. I recommend the paper be accepted for publication.

Response: We thank the reviewer for their encouraging feedback, no specific revisions were requested.

Reviewer #2 (Remarks to the Author):

Summary

The manuscript describes the use of an energy selecting filter combined with electron counting detector for the collection of electron diffraction data from lamellae of proteinase K. This builds on previous work by the authors demonstrating the use of electron counting detectors for electron diffraction. The manuscript argues and concludes that the use of the energy filter improves the overall resolution of the data by filtering out the inelastically scattered electrons. Evidence for this is provided in the form of example diffraction images with specific reflections highlighted with pixel intensity plots. The manuscript also provides multiple processing statistics from a number of data collections using the energy filter. Example electrostatic potential maps are provided to demonstrate the quality of the data and difference maps are provided to demonstrate that the data includes density for hydrogen positions. There is a limited comparison of the energy filtered data with the unfiltered data, particularly comparing the data quality indicators to those from previously published work (Martynowycz, M. W. et al. Nat. Commun. 14, 1086 (2023)).

The authors refer to work by Yonekura et al. (Yonekura, K., Ishikawa, T. & Maki-Yonekura, S. A new cryo-EM system for electron 3D crystallography by eEFD. J. Struct. Biol. 206, 243–253 (2019)) and state that their work does not demonstrate any increase in resolution due to the use of energy filter. This is true because the resolution of their experiments was limited by detector geometry. However, the data quality statistics indicate clearly that the signal to noise of the data is substantially improved (by about a factor of 3 in both their test cases at the highest measured resolution) where energy filtering is employed. For completeness, this point should be mentioned by the authors of this manuscript.

Response: We made the changes as suggested. This part of the introduction on page 2, lines 22-26 now reads: “*Previous reports assessed the effects of energy filtering on macromolecular diffraction data, demonstrating a significant reduction in background noise and sharper spots, resulting in improved data quality statistics^{3,12,13}. Owing to detector geometry, resolution did not improve even though the signal-to-noise ratio increased substantially¹³.*”

While the manuscript demonstrates an increase in attainable resolution through the use of an energy filter combined with electron counting detector, it demonstrates this on one well diffracting system and on measurements recorded from different sample crystals. Overall, in its current state the manuscript lacks the sufficient detail and a robust comparison with non-energy filtered data to support the conclusions made. Specifically, it is unclear which ‘unfiltered’ data sets were used for the comparison and precisely how they were prepared, which lamella thicknesses were used, how many crystals were merged, and so on. At very least a side-by-side comparison table between unfiltered and filtered data should be included detailing this information to allow the reader to judge their interpretation.

Response: For comparison, we used unfiltered MicroED data acquired with the same experimental setup, previously reported in Martynowycz *et al.*, Nature Communications 14, 2023. The lamellae were machined to 300 nm thickness, and data of several crystal lamellae (five for unfiltered, 17 for filtered datasets) were merged. We have clarified this in the main text on page 3, line 3, and added **Extended Data Table 2**, which provides the side-by-side comparison of filtered and unfiltered MicroED data.

An ideal experimental comparison would have been to record filtered and unfiltered from the same lamella, thereby reducing impact of crystal-to-crystal variation. Was this considered and tried? The reviewers suggest that this may be a more rigorous demonstration of any observed gains in signal to noise.

Response: While we agree that an in-depth comparison of unfiltered and filtered data would offer valuable insight, such an approach is rather challenging: sequentially recording both filtered and unfiltered data from the same lamella will reduce differences due to crystallinity and non-isomorphism, but the accumulated dose, and consequently radiation damage, will differ significantly. High-resolution spots often start to fade after the first initial frames. Addressing the dose discrepancy would require interleaving data collection for the two datasets, which is not possible under continuous rotation. We deem these complicated measurements to be beyond the scope of our original short report. Our study is aimed at demonstrating the full potential of energy filtering combined with electron counting, illustrating what can be achieved with resolution and data quality using this strategy. We believe these results, along with the new side-by-side comparison in **Extended Data Table 2**, effectively highlight the key improvements enabled by energy filtering.

The manuscript is of interest to the emerging community of electron diffraction users and facilities and represents a potential incremental but important improvement to data quality. To support their conclusions, we suggest that the authors consider the suggestions made above and also consider a new set of experimental data collections.

Detailed Comments

- Are the distances appropriate for the positive densities in figure 2 to be representative of protons?

Response: The distances presented in Figure 2 are measured between non-hydrogen atoms, and are consistent with interatomic distances observed in high-resolution X-ray structures. We have updated these panels in **Figure 2** to enhance clarity, making the distances more evident.

- Given that some fraction of the inelastically scattered electrons contribute to the Bragg peaks (i.e. elastic event followed by inelastic event), and contribute to measurable and potentially useful signal, what impact does this have on the conclusions drawn?

Response: While it is true that some inelastically scattered electrons, particularly those that have lost a small fraction of their energy (e.g., ~20 eV out of 300 keV), may retain partial coherence and thus could contribute a measurable signal, these electrons still experience a phase shift. This phase shift, though small, diminishes the structural information they carry compared to fully coherent elastic scattering events. Importantly, as the energy loss increases (hundreds or thousands of eV), these electrons no longer contribute meaningfully to the diffraction pattern and instead add to the background, which can obscure or degrade the overall signal.

Given that we cannot easily separate electrons that have lost small amounts of energy (which may still contribute useful information) from those that have undergone significant energy loss, the prudent approach is to filter out inelastically scattered electrons. By using energy filters to remove electrons that have lost a specific threshold of energy, we ensure that the signal derived from elastic scattering dominates, eliminating interference and ambiguity. This approach enhances the signal quality, isolates the most coherent electron contributions, and strengthens the overall conclusions drawn from the diffraction data.

We added additional discussion on page 3, lines 9-12: *“Although inelastically scattered electrons with minimal energy loss may contribute some useful signal, their phase shift reduces structural coherence, and those with significant energy loss primarily add to background noise. The most coherent electron contributions can be isolated using energy filtering, thus enhancing the signal and data quality.”*

- What were the last diffraction weighted doses? The manuscript refers to low electron fluence, but it would be good to understand the impact as dose. These can now be easily calculated using Raddose-ED. Particularly to be able to compare the required doses needed for filtered Vs un-filtered for equivalent diffraction quality.

Response: The flux density, fluence, and dose values have been included in the new **Extended Data Table 2** for comparison.

- The extended data in the manuscript depicts diffraction using 2 sizes of energy selecting slit but no comment or comparison is made on the effect of using the narrower slit.

Response: To maintain focus, we removed the additional figure from the manuscript.

- The manuscript only demonstrates a single crystal system. The manuscript would be strengthened by the demonstration of the technique on a different crystal system where the unit cell size, spacegroup and protein mass are reasonably different.

Response: Proteinase K is a well-characterized standard sample that is very close to the median size for a protein from the human proteome. Proteinase K is therefore routinely used for evaluating MicroED

methods due to its reproduceable and suitability for benchmarking. While other crystal systems could be explored, doing so would add unnecessary complexity without providing additional insights into the performance of energy filtering. We used a reliable model system to demonstrate the effectiveness of energy filtering and electron counting.

- What was the distribution of the combined datasets in terms of resolution range, were all the images used? Were all the images used from each individual dataset?

Response: Individual datasets ranged from 1.13 to 1.06 Å resolution, we clarified this in the main text on page 2, line 41. All images were used for each individual dataset, which is now mentioned on page 6, line 17. We also clarified this in the updated caption of **Extended Data Figure 4**.

- What do the statistics look like for an equivalent 17 merged datasets from non-energy filtered conditions in the same system?

Response: We included **Extended Data Table 2**, which provides a side-by-side comparison of the data processing and refinement statistics for the filtered and unfiltered MicroED data.

- Figure 2A. The comparison of intensity statistics for filtered Vs unfiltered. Unclear if these are merged or single data sets.

Response: Both the filtered and unfiltered data are merged from multiple datasets. This is made clear in the new **Extended Data Table 2**.

- Extended figure 4 – CC1/2 please add dotted line denoting the resolution cut-off on for CC1/2 axis.

Response: We added dotted lines for the cut-off.

- The comparable data between filtered and unfiltered in the manuscript is limited, it extends only to diffraction images and a single comparison of intensity, CC1/2 and Rpim statistics and it is not clear how reproducible or robust these are.

Response: We have included **Extended Data Table 2** enabling a full comparison of intensity and model statistics between filtered and unfiltered data.

- Please include unfiltered data in extended data table 1 so that comparison of statistics can be made. E.g. there are no values for unfiltered obs. or unique obs. and so the associated statement in the text cannot be backed up.

Response: We have included these values in **Extended Data Table 2**.

- Can the background signal be quantified for energy filtered vs unfiltered?

Response: We calculated radial averages of the diffraction frames to quantify the background signal of the filtered data shown in **Figure 1** versus the unfiltered data presented in **Extended Data Figure 1**. We

added these plots as new **Extended Data Figure 2** to enable a comparison, and refer to the plots in the main text on page 2, lines 37-38

- It is not clear what extended figure 2 offers in addition to figure 1. Which energy filter was used to generate figure 1?

Response: **Extended Data Figure 2** shows an initial 5s frame from a small wedge MicroED movie of a rotating crystal. The total fluence over this frame was $0.01 \text{ e}^-/\text{\AA}^2$. **Figure 1** shows a still diffraction pattern of a stationary crystal with a total fluence of $0.84 \text{ e}^-/\text{\AA}^2$ collected over a 420 second exposure. We have clarified this and added the corresponding exposure times and fluence for each frame in the updated captions to **Figure 1** and **Extended Data Figures 1** and **3**.

- Figure 2. A. “Comparison of intensity statics for ...” I think should read “Comparison of intensity statistics for ...”

Response: We fixed this in the caption of **Figure 2**.

- Did the refined detector distance correlate with the calibrated distance and was this the same for all datasets?

Response: The detector distance for all data was calibrated using a standard evaporated aluminum grid at 1402 mm, which is stated in the Methods section page 6, lines 1-2. The distance was not explicitly refined during data processing, which is now stated in the revised manuscript on page 6, lines 15-16.

- What were the approximate/average lamella dimensions? Were the lamellae narrower than the parallel electron beam?

Response: The lamellae were machined to 300 nm thickness and 5 μm in width. This is now stated on page 5, line 25. The crystalline parts of the lamellae were larger than the parallel electron beam with a diameter of 3.5 μm at the sample plane as defined by the SA aperture.

- The manuscript should include data processing scripts in the supplementary information.

Response: Each dataset was processed manually using XDS. No script was used.

Reviewer #3 (Remarks to the Author):

In this manuscript, the authors demonstrated a significant advancement in MicroED by utilizing a post-column energy filter and a direct electron detector for data collection. These innovations substantially improved the signal-to-noise ratio in electron diffraction patterns collected from protein crystals. This technical breakthrough enabled the sub-atomic resolution structure determination of proteinase K crystals by MicroED, representing a considerable update to the state-of-the-art in macromolecular studies using MicroED.

In recent years, the authors' group has pioneered several techniques to enhance the capability of MicroED. Their contributions include optimizing cryo-FIB protocols, including integrating Correlative Light and Electron Microscopy (CLEM) and plasma-FIB for the preparation of frozen hydrated crystal lamellae. They also

demonstrated it was ideal to employ direct electron detectors for ultra-low-dose and high resolution MicroED data collection. In this work, they further refined their approach by incorporating an energy filter to eliminate inelastically scattered electrons from diffraction patterns, thereby enhancing the data precision and accuracy. As a result, they improved the resolution of tetragonal proteinase K from 1.4 Å (Martynowycz et al., Nature Communications 2023) to an impressive 1.09 Å. This achievement marks a significant milestone in electron crystallography method development and will undoubtedly open new avenues in structural biology.

The experiments are well-conceived and executed, and the manuscript is written clearly. I therefore recommend this manuscript for publication in Nature Communications. However, there are a few points that I would like the authors to address:

- By comparing the diffraction patterns in Figure 1 and Figure S1, it is evident that the energy filter enhanced the signal-to-noise ratio. While background noise in the DED remains comparable (both very dark), inelastically scattered electrons lose energy during scattering, causing a slight deviation from the Bragg position and creating a ‘halo’ around elastically scattered electrons. This results in reduced peak intensity and broader peak profiles in the non-filtered case. The use of the energy filter not only improves low-resolution data (Yonekura, *PNAS* 2015) but also makes high-resolution, low-intensity peaks more discernible.

On the other hand, the authors used a small-wedge data collection strategy, deviating from the ultra-low-dose, high-tilt protocol employed in their previous work. The increased electron dose per frame could also have contributed to the improved signal-to-noise ratio, though this aspect was not emphasized in the manuscript.

Response: The Conclusion was amended with a sentence on the contribution of the smaller wedge data collection scheme to the signal-to-noise ratio. The added sentence on page 3, line 20 reads: “*Our optimized protocol, which utilizes small wedges and therefore allows for an increased flux per slice through reciprocal space, may have further contributed to the improved signal-to-noise ratio.*”

- Since data from 17 crystals were scaled and merged, redundancy increased compared to earlier studies by the authors. Did this increase in redundancy contribute to the enhancement of the electrostatic potential and the overall model quality?

Response: We have added **Extended Data Table 2** to allow a more detailed comparison with our previous unfiltered MicroED data. The overall redundancy of the merged data from 17 crystals was 28.6, compared to 25.8 for the unfiltered data, which were obtained by merging 5 crystals. While the slightly higher redundancy may have contributed positively to the map and model quality, any potential negative effects of non-isomorphism across a larger set of crystals cannot be ruled out.

- It would be valuable if the authors could compare the structural models and electrostatic potential maps from this study with those from their previous work. Did the relatively higher dose rate result in any visible beam damage?

Response: The total fluence used for each crystal dataset in this study was $0.84 \text{ e}^-/\text{Å}^2$, slightly lower than the $1.0 \text{ e}^-/\text{Å}^2$ fluence used in previous unfiltered experiments. We added these values to **Extended Data Table 2** for comparison. Visual inspection of the maps did not reveal any apparent signs of radiation

damage, expect for potentially residue Glu132. This residue appears rather poorly resolved in both maps, which could indicate radiation damage to its carboxyl group, which can be more susceptible to beam damage in cryo-EM. Alternatively, this residue might be more flexible as it is located on the outside of the protein exposed to the bulk solvent.

- The authors mention that “filtering out the noise revealed diffuse scattering phenomena that could hold additional structural information.” Could they elaborate on what new structural insights were revealed, especially in comparison to their earlier studies? How does this new information compare with X-ray structures?

Response: In our data, the observed diffuse scattering features are characteristic of the bulk solvent, with distinct rings at approximately 3.5 Å and 2.2 Å resolution. While these specific patterns are not highly informative on their own, previous studies in macromolecular X-ray crystallography and electron diffraction of materials have shown that diffuse scattering can indicate conformational changes or structural disorder. Future work may explore these phenomena in greater detail to assess their relevance for structural insights into macromolecular systems.

- The best X-ray Proteinase K structure cited (PDB 5KXV, Masuda et al., Sci. Rep. 2017) may not be the most ideal, for example PDB 7LTD was higher in resolution. I hope the authors could conduct a thorough search. While I acknowledge that MicroED data provides richer information than X-ray data, an interesting observation is that despite comparable resolutions, X-ray electron density maps resolve atomic positions more clearly. This highlights the need for more accurate electron scattering factors to improve electrostatic potential modeling.

Blue – MicroED, Pink – X-ray, both contoured at 3 RMSD.

Response: We added a citation for Yabukarski *et al.*, Acta Cryst. D78 (2022) referencing PDB structure 7TLD. We agree that more accurate electron scattering factors would improve the modelling of

electrostatic potential maps in electron diffraction. We added the following comment to the Conclusion on page 3, lines 17-19: “*At comparable resolution, the atomic positions are more clearly resolved in the electron density maps, highlighting the need for more accurate electron scattering factors and improved modeling of the electrostatic potential distribution.*”

- Certain regions of the model could be further improved, such as the conformation of GLU 132, the alternative conformations of the CYS 139 – CYS 228 disulfide bond, and water.

Response: We have updated the coordinates as suggested. Several waters with close contacts, along with one water in a special position, were removed from the model. An alternative conformation of the Cys139-Cys228 was modeled. Regarding the previous comments on beam damage: Glu132 was added in alternate conformations, although this residue remained poorly resolved likely indicating either radiation damage to the carboxyl group or increased flexibility due to solvent exposure. Following these adjustments, the structure was refined anew, and the updated coordinates and structure factors were submitted to the Protein Data Bank with accession code 9DHO. A validation report along with the revised coordinate and structure factor files is included with the resubmission.

Reviewer #4 (Remarks to the Author):

Response: We thank the reviewer for their efforts and valuable input in evaluating our manuscript.

Yours sincerely,

Tamir Gonen, on behalf of the authors

Response to Reviewer Comments:

Reviewer #2 (Remarks to the Author):

We would to thank the authors for their work in amending the manuscript.

The manuscript has been substantially improved with the introduction of a clearer side-by-side comparison of the unfiltered and filtered data sets. Our overriding concern still is that the significantly wider sweeps of data per crystal (63 degs. vs 20 degs.) may well mean the unfiltered data were more impacted by beam damage. i.e. the resolution at the start of the data collection may have been higher than that at the end of data collection and therefore higher than the final resolution cut off. All of the other concerns have been addressed.

We suggest that the authors check the data reduction scale factors of both filtered and unfiltered data sets and quantify the beam damage effects (looking at the scaling B factors for example, or loss of mean intensity at a given resolution) for both filtered and unfiltered sets. This would be to confirm that beam damage in the unfiltered data sets does not contribute substantially to the differing data quality of filtered vs unfiltered data.

Response: We appreciate the reviewer's concerns about the potential impact on the attainable resolution for the wider 63° sweeps used for the unfiltered data compared to the 20° wedges used for the filtered data. Beam damage during data collection will cause the mean intensity of reflections to decrease and leads to a loss of high-resolution information. However, the total fluence, which is linearly correlated to absorbed dose, was similar for both data collection strategies, $1.0 \text{ e}^-/\text{Å}^2$ (3.7 MGy) for the unfiltered data versus $0.84 \text{ e}^-/\text{Å}^2$ (3.1 MGy) for the filtered 20° sweeps. These differences are relatively small, and the beam damage would affect the decay in intensities similarly in either strategy. For example, we do see a decrease in attainable resolution in the filtered data where on the first frames we observe spots at 0.97 Å resolution (**Extended Data Figure 3**), but we only are able to integrate data reliably up to 1.06 Å resolution. An in-depth analysis of beam damage, is not trivial and beyond the scope of the current manuscript which is focused on energy filtering. Importantly, the effects of beam damage on MicroED data quality, intensity decay, and *B*-factors have been extensively discussed for proteinase K in Hattne *et al.*, *Structure* **26**, 2018.

Whereas beam damage would increase identically with absorbed dose in either data collection protocol, the higher fluence per wedge for the 20° sweeps might have contributed to a better signal-to-noise ratio. To further address these concerns about the impact of the sweep angles, we conducted an additional experiment to compare equivalent unfiltered MicroED data that were collected using 20° sweeps. Data from 12 crystals were collected identically to the filtered MicroED data, with the exception that the energy filter slit was retracted. These results are now included in **Extended Data Table 2**. **Notably, this comparison shows no significant improvement in data quality or resolution over the 63° sweeps, reinforcing our previous conclusion that the observed improvements in data quality and attainable resolution are primarily attributable to energy filtering.**

We added the following lines to the main text, page 3, line 4: “*Additionally, we collected unfiltered data using the same 20° sweep strategy described here, which did not yield any improvement in data quality over the previous unfiltered data from the larger sweeps (Extended Data Table 2).*” Furthermore, we expanded our discussion on the different data collection sweeps in the main text, page 3, line 24. This section now reads: “*Our optimized protocol, which utilizes small 20° sweeps and therefore allows for an increased flux per slice through reciprocal space, may have further contributed to the improved signal-to-noise ratio. Interestingly, the smaller sweeps alone did not improve the unfiltered data compared to the larger 63° sweeps used previously, indicating that the improvements in data quality and resolution can mainly be attributed to the removal of the inelastic scattering*

contributions. The decay in mean diffracted intensity and subsequent loss of high-resolution information can be expected to occur identically in either of the data collection strategies as both resulted in a similar dose¹⁸.”

Comments

1. In the Data Collection section, it would be very helpful to explicitly include the equivalent data collection parameters used for the original unfiltered data sets. These could possibly be included in a Table for clarity.

Response: We added an additional paragraph titled “*Data comparison*” to the Methods section detailing the two unfiltered datasets that were added for comparison in the updated **Extended Data Table 2**. This entire section reads: “*The energy-filtered MicroED data were compared to two unfiltered datasets that were collected using the same experimental setup: The first set was obtained by merging unfiltered data from five proteinase K lamellae using 63.0° sweeps at a total fluence of $\sim 1.0 \text{ e}^-/\text{Å}^2$ (3.7 MGy) and was previously reported in Martynowycz et al., 2023. The second set merged unfiltered data from 12 lamellae using the same 20.0° sweep data collection strategy as described for the filtered MicroED data, where each dataset was recorded using a total fluence of $\sim 0.84 \text{ e}^-/\text{Å}^2$ (3.1 MGy) with the energy filter slit retracted. The merged data were truncated at 1.4 Å resolution to enable an equal comparison of the intensity and model statistics.*”

2. Figure 1 legend. “... over a 420 s expose at a total ...” should read “... over a 420 s exposure at a total ...”

Response: We made the changes as suggested.

Reviewer #3 (Remarks to the Author):

I appreciate the authors' efforts in addressing the reviewers' comments and improving the manuscript. I am pleased with the revisions and fully support its publication in Nature Communications.

I would like to reiterate that the improved structure determination is not solely due to the use of the energy filter, but also to the updated small-wedge data collection strategy. I believe the higher fluence per frame contributed to the improved signal-to-noise ratio as well (newly added Extended Data Table 2). I appreciate that the authors included a statement on this in the revised manuscript.

Response: We thank the reviewer for their comments. As per response to reviewer #2, we added a brief discussion on the effects of wedge sizes and fluence to the main text, page 3, lines 4-6 and 26-30.

Reviewer #4 (Remarks to the Author):

Response: We thank the reviewer for their efforts. No further changes were requested.

In this manuscript, the authors demonstrated a significant advancement in MicroED by utilizing a post-column energy filter and a direct electron detector for data collection. These innovations substantially improved the signal-to-noise ratio in electron diffraction patterns collected from protein crystals. This technical breakthrough enabled the sub-atomic resolution structure determination of proteinase K crystals by MicroED, representing a considerable update to the state-of-the-art in macromolecular studies using MicroED.

In recent years, the authors' group has pioneered several techniques to enhance the capability of MicroED. Their contributions include optimizing cryo-FIB protocols, including integrating Correlative Light and Electron Microscopy (CLEM) and plasma-FIB for the preparation of frozen hydrated crystal lamellae. They also demonstrated it was ideal to employ direct electron detectors for ultra-low-dose and high resolution MicroED data collection. In this work, they further refined their approach by incorporating an energy filter to eliminate inelastically scattered electrons from diffraction patterns, thereby enhancing the data precision and accuracy. As a result, they improved the resolution of tetragonal proteinase K from 1.4 Å (Martynowycz et al., *Nature Communications* 2023) to an impressive 1.09 Å. This achievement marks a significant milestone in electron crystallography method development and will undoubtedly open new avenues in structural biology.

The experiments are well-conceived and executed, and the manuscript is written clearly. I therefore recommend this manuscript for publication in *Nature Communications*. However, there are a few points that I would like the authors to address:

- By comparing the diffraction patterns in Figure 1 and Figure S1, it is evident that the energy filter enhanced the signal-to-noise ratio. While background noise in the DED remains comparable (both very dark), inelastically scattered electrons lose energy during scattering, causing a slight deviation from the Bragg position and creating a 'halo' around elastically scattered electrons. This results in reduced peak intensity and broader peak profiles in the non-filtered case. The use of the energy filter not only improves low-resolution data (Yonekura, *PNAS* 2015) but also makes high-resolution, low-intensity peaks more discernible.

On the other hand, the authors used a small-wedge data collection strategy, deviating from the ultra-low-dose, high-tilt protocol employed in their previous work. The increased electron dose per frame could also have contributed to the improved signal-to-noise ratio, though this aspect was not emphasized in the manuscript.

- Since data from 17 crystals were scaled and merged, redundancy increased compared to earlier studies by the authors. Did this increase in redundancy contribute to the enhancement of the electrostatic potential and the overall model quality?
- It would be valuable if the authors could compare the structural models and electrostatic potential maps from this study with those from their previous work. Did the relatively higher dose rate result in any visible beam damage?

- The authors mention that “filtering out the noise revealed diffuse scattering phenomena that could hold additional structural information.” Could they elaborate on what new structural insights were revealed, especially in comparison to their earlier studies? How does this new information compare with X-ray structures?
- The best X-ray Proteinase K structure cited (PDB 5KXV, Masuda et al., Sci. Rep. 2017) may not be the most ideal, for example PDB 7LTD was higher in resolution. I hope the authors could conduct a thorough search. While I acknowledge that MicroED data provides richer information than X-ray data, an interesting observation is that despite comparable resolutions, X-ray electron density maps resolve atomic positions more clearly. This highlights the need for more accurate electron scattering factors to improve electrostatic potential modeling.

Blue – MicroED, Pink – X-ray, both contoured at 3 RMSD.

- Certain regions of the model could be further improved, such as the conformation of GLU 132, the alternative conformations of the CYS 139 – CYS 228 disulfide bond, and water.